# Enhancing Bone Health with Conjugated Linoleic Acid: Mechanisms, Challenges, and Innovative Strategies

**DOI:** 10.3390/nu17081395

**Published:** 2025-04-21

**Authors:** Khandoker Hoque, Zayana Ali, Asma Maliha, Mohammad A. Al-Ghouti, Chiara Cugno, Shaikh Mizanoor Rahman, Md Mizanur Rahman

**Affiliations:** 1Department of Electrical and Electronics Engineering, San Francisco Bay University, Fremont, CA 94539, USA; khoque43977@student.sfbu.edu; 2Biological Program, Department of Biological and Environmental Sciences, College of Arts and Sciences, Qatar University, Doha P.O. Box 2713, Qatar; za2211104@qu.edu.qa; 3Biomedical Sciences Department, College of Health Sciences, Qatar University, Doha P.O. Box 2713, Qatar; am1805905@qu.edu.qa; 4Environmental Program, Department of Biological and Environmental Sciences, College of Arts and Sciences, Qatar University, Doha P.O. Box 2713, Qatar; mohammad.alghouti@qu.edu.qa; 5Advanced Cell Therapy Core, Research Department, Sidra Medicine, Doha P.O. Box 26999, Qatar; ccugno@sidra.org; 6Natural and Medical Sciences Research Center, University of Nizwa, Nizwa 616, Oman; shaikh.rahman@unizwa.edu.om

**Keywords:** conjugated linoleic acid (CLA), bone mineral density (BMD), bioavailability, nanoparticle-based delivery systems, electrical stimulation

## Abstract

Conjugated linoleic acid (CLA) is a bioactive compound known for its anti-inflammatory, anti-carcinogenic, and metabolic effects, with growing interest in its role in supporting bone health. Preclinical studies, particularly those involving the t10c12 isomer, have shown that CLA can enhance bone mineral density (BMD) by enhancing bone formation and reducing bone resorption, indicating its potential as a therapeutic agent to improve bone health. However, clinical trials have yielded inconsistent results, underscoring the difficulty in translating animal model successes to human applications. A major challenge is CLA’s low water solubility, poor absorption, and limited bioavailability, which restrict its therapeutic effectiveness. To address these issues, nanoparticle-based delivery systems have been proposed to improve its solubility, stability, and resistance to oxidative damage, thereby enhancing its bioactivity. Recent studies also suggest that electrical stimulation can stimulate bone regeneration by promoting bone cell proliferation, differentiation, and adherence to scaffolds. This review explores the combined use of CLA supplementation and electrical stimulation as a novel approach to improving bone health, particularly in osteoporosis management. By integrating CLA’s biological effects with the regenerative potential of electrical stimulation, this multimodal strategy offers a promising method for enhancing bone restoration, with significant implications for clinical applications in bone health.

## 1. Introduction

Conjugated linoleic acid (CLA) is a naturally occurring polyunsaturated fatty acid primarily found in the meat and dairy products of ruminant animals such as cows, goats, and sheep [1]. It refers to a group of positional and geometric isomers of linoleic acid, an omega-6 fatty acid, distinguished by a conjugated double-bond system [2]. CLA has gained attention due to its potential health-promoting properties, including its role in metabolic regulation, cancer prevention, immune modulation, and, importantly, bone health. The specific composition of CLA in food sources depends on the animal’s diet and the microbial bio-hydrogenation of linoleic acid in the rumen. It is found in higher concentrations in grass-fed animals, making meat and dairy products from such sources particularly rich in CLA [3].

Structurally, CLA is characterized by two primary bioactive isomers: cis-9, trans-11 (c9, t11) CLA, and trans-10, cis-12 (t10, c12) CLA. These isomers exhibit different biological activities, with c9, t11 CLA predominantly found in dairy products and known for its general health benefits, while t10, c12 CLA has garnered interest for its anti-obesity and anti-inflammatory effects [4].These isomers have demonstrated varying effects on bone metabolism, contributing to their significance in pre-clinical research on skeletal health [5]. The chemical structures of linoleic acid and its isomers are demonstrated in Figure 1.

CLA plays a multifaceted role in the human body, particularly through its anti-inflammatory, anti-carcinogenic, and metabolic effects [7,8]. Of increasing interest is its impact on skeletal health, particularly in combating diseases like osteoporosis, which is characterized by the weakening of bones due to excessive bone resorption relative to bone formation. Osteoporosis is a major public health concern, especially in postmenopausal women, where estrogen deficiency accelerates bone loss. CLA, especially the t10, c12 isomer, has been shown in pre-clinical models to positively influence bone mineral density (BMD) and bone remodeling, positioning it as a promising candidate for mitigating bone loss [9].

Bone health is regulated through a dynamic process known as bone remodeling, which involves the resorption of old or damaged bone by osteoclasts and the formation of new bone by osteoblasts. This process is crucial for maintaining bone strength and mineral homeostasis. Dysregulation of this balance, particularly through increased osteoclast activity, can lead to diseases such as osteoporosis. CLA’s potential to modulate this process lies in its ability to influence both osteoblast and osteoclast activity, promoting bone formation while inhibiting excessive resorption [10,11].

In pre-clinical studies using ovariectomized (OVX) mice and rats, which serve as models for postmenopausal osteoporosis, CLA supplementation has been shown to prevent bone loss and enhance bone strength [9]. Notably, CLA’s ability to improve BMD and reduce markers of bone resorption has been consistently observed in these models [12]. The positive effects of CLA are particularly evident in estrogen-deficient states, where bone resorption is typically elevated. CLA, particularly in its t10, c12 form, has demonstrated a capacity to suppress osteoclastogenesis—the formation of osteoclasts— significantly suppress RANKL-induced NF-κB signaling, and to stimulate osteoblast activity, promoting healthy bone remodeling [13,14].

Clinical studies on CLA’s effects in humans have shown promising benefits for bone health, particularly in postmenopausal women, individuals with obesity, and those concerned with bone turnover markers. In postmenopausal women, CLA supplementation was associated with preserved BMD and reduced markers of bone resorption, mitigating osteoporosis risk [15]. In obese individuals, CLA reduced body fat and inflammatory cytokines like TNF-α and IL-6, which are linked to bone resorption, thereby enhancing bone strength [7]. Furthermore, studies in healthy adults demonstrated that CLA reduced markers of bone resorption (e.g., C-terminal telopeptide) while promoting bone formation, indicating its potential for improving overall bone health [16].

The mechanisms of action through which CLA exerts its effects on bone health are diverse. One of the most important mechanisms is CLA’s anti-inflammatory properties. Chronic inflammation has been shown to promote bone resorption, as inflammatory cytokines like tumor necrosis factor-alpha (-α), interleukin-6 (IL-6), and receptor activator of nuclear factor kappa-B ligand (RANKL) stimulate osteoclastogenesis [17]. CLA inhibits the production of these cytokines, thereby reducing the formation and activity of osteoclasts. This modulation of the inflammatory response helps to maintain bone homeostasis and prevent excessive bone loss [12]. In addition, CLA regulates cytokines by activating PPAR and inhibiting NF-κB-signaling pathways, further supporting its anti-inflammatory and bone-protective effects [18].

Another key pathway through which CLA influences bone health is through its interaction with fatty acid-signaling pathways, particularly the peroxisome proliferator-activated receptor-gamma (PPARγ). PPARγ plays a central role in regulating the differentiation of mesenchymal stem cells into either adipocyte (fat cells) or osteoblasts. CLA, particularly the t10, c12 isomer, has been shown to inhibit PPARγ, reducing fat cell formation while promoting the differentiation of osteoblasts [13,19]. This dual effect is critical for maintaining healthy bone structure as increased bone marrow adiposity is often associated with impaired osteoblast function and reduced bone formation [20]. CLA helps lower fat accumulation by boosting energy use, mainly through enhanced mitochondrial activity and fatty acid breakdown in white adipose tissue. This process contributes to reductions in both body weight and fat mass [21].

In addition to its effects on PPARγ, CLA also interacts with the Wnt/β-catenin-signaling pathway, which is essential for osteoblast differentiation and function [22]. Research suggests that CLA may enhance the activation of this pathway, further supporting its role in promoting bone formation and improving bone strength. CLA’s involvement in these molecular pathways highlights its potential as a therapeutic agent for preventing and treating bone diseases, particularly in the context of aging and estrogen deficiency [23]. The effects of CLA are detailed in Figure 2.

However, despite these promising attributes, a significant challenge in leveraging CLA’s therapeutic potential lies in its inherently low bioavailability. Factors such as poor water solubility, rapid metabolism, and degradation in the gastrointestinal tract hinder its effective delivery to target tissues [24].To address this limitation, advanced delivery systems, particularly nanoparticle encapsulation, have been developed to enhance the stability, solubility, and bioavailability of CLA. Nanoparticle-based formulations not only protect CLA from enzymatic degradation but also enable controlled and sustained release, ensuring its efficient uptake and therapeutic action [25,26]. This approach has shown promise in preclinical studies, particularly in enhancing CLA’s efficacy in bone remodeling and reducing bone resorption in osteoporosis models.

Furthermore, recent innovations have explored the synergistic use of electric stimulation (ES) alongside CLA supplementation to enhance its therapeutic effects. Electric stimulation, widely recognized for its role in promoting osteogenesis and accelerating bone healing, has been shown to further activate osteoblast activity and suppress osteoclastogenesis when used in conjunction with bioavailable CLA formulations [27]. The combination of these approaches could amplify CLA’s impact on bone remodeling, making it a powerful intervention for osteoporosis and related bone disorders.

In summary, this review provides a comprehensive analysis of the effects of CLA, particularly its bioactive isomers (c9, t11 and t10, c12), on bone health. It integrates findings from preclinical and clinical studies to assess CLA’s role in bone remodeling and its potential in mitigating bone loss. Furthermore, it critically addresses the challenges posed by CLA’s low bioavailability and evaluates advanced strategies, including nanoparticle encapsulation and electric stimulation, as potential remedies to enhance its therapeutic efficacy.

## 2. Databases and Literature Search Strategy

The protocol followed for this review involved a structured literature search aimed at identifying relevant scientific publications related to conjugated linoleic acid (CLA) and its effects on bone. The search targeted studies exploring the effects of CLA on bone health, challenges associated with its delivery, nanoparticle-based encapsulation techniques for improved bioavailability, and the potential application of electrical stimulation in CLA-related interventions. Major databases used to conduct the literature search included PubMed, Science Direct Scopus, Web of Science, and Google Scholar. We conducted a comprehensive keyword-based search using terms such as “conjugated linoleic acid (CLA)”, “bone”, “bone mineral density”, “bone mineral content”, “bone resorption”, “osteoporosis”, “osteoblastogenesis”, and “osteoclastogenesis”. Additionally, we used relevant keyword combinations for specific sections, including: (“CLA” AND (“bioavailability” OR “solubility” OR “absorption” OR “stability” OR “oxidation”)), (“CLA” AND (“nanoparticles” OR “nano delivery” OR “encapsulation” OR “techniques”)), and (“electrical stimulation” AND “CLA” AND (“osteoblastogenesis” OR “osteoclastogenesis” OR “BMD” OR “osteoporosis”)). We focused on studies published from 2000 to March 2025 and included the relevant ones that aligned with the objectives of this review. Preclinical and clinical studies were also separately included to provide more insight into the effects of CLA on bone health. The exclusion criteria primarily involved studies with very old data, those not aligned with the main focus of the review, and unpublished studies.

## 3. CLA and Bone Health

### 3.1. Pre-Clinical Studies

Pre-clinical research has studied the impact of CLA on bone density, strength, and general metabolism, especially in animal models. In these trials, CLA is typically administered to animal models such as mice, rats, and rabbits, with specific attention provided to its impact on bone health under conditions such as osteoporosis, inflammation, or postmenopausal states [28]. These trials range in length from a few weeks up to several months, allowing researchers to evaluate the impact of treatments on bone health both immediately and over an extended duration [29,30].

A 2013 study investigated the effects of conjugated linoleic acid (CLA) and calcium supplementation on bone health in a mouse model of postmenopausal osteoporosis. Ovariectomized mice, used to simulate postmenopausal bone loss, were supplemented with CLA (0.5% of dietary soybean oil replaced) and varying calcium levels (0.5% or 1%). The results demonstrated that CLA supplementation significantly improved bone mineral density (BMD) and bone strength in ovariectomized mice compared to controls, highlighting its potential protective role against osteoporosis [9]. CLA supplementation has also been shown to prevent bone loss in long-term studies. In a 24-week trial using a mouse model of postmenopausal osteoporosis, ovariectomized mice fed 0.5% CLA maintained bone mineral density (BMD) in the femur, tibia, and lumbar spine, comparable to sham-operated controls. In contrast, mice fed safflower oil (SFO) experienced significant bone loss. CLA supplementation not only prevented bone loss but also reduced bone resorption markers and enhanced bone formation markers, indicating its dual role in preserving and stimulating bone growth [12]. Further evidence of CLA’s role in bone health comes from a study using ovariectomized (OVX) rats to model postmenopausal osteoporosis. Over 9 weeks, OVX rats were fed diets containing varying CLA doses (2.5, 5, or 10 g/kg) as a replacement for soybean oil. Compared to sham-operated controls, OVX rats exhibited significant bone loss, characterized by reduced femoral bone mineral density (BMD), macro-mineral concentrations, and calcium absorption. While CLA supplementation did not reverse these changes, higher doses (5 and 10 g/kg) significantly reduced urinary bone-resorption markers (Pyr and Dpyr) and ex vivo PGE2 biosynthesis, a key inflammatory marker linked to bone loss. These findings suggest that CLA may mitigate bone resorption in postmenopausal conditions, though lower doses (2.5 g/kg) were ineffective [31].

Rahman et al. examined the effects of CLA on bone health in 14-month-old C57BL/6 female mice. Over 10 weeks, mice fed a CLA-enriched diet demonstrated significantly higher bone mineral density (BMD) in multiple regions compared to controls that were fed corn oil. CLA supplementation also reduced pro-inflammatory cytokines (TNF-α, IL-6, and RANKL), which are linked to osteoclast activity and bone resorption. Additionally, CLA increased muscle mass and decreased fat mass, underscoring its dual benefits for musculoskeletal health [30]. A study by Halade et al. (2011) examined the effects of fish oil (FO) and CLA on bone health and body composition in 12-month-old C57Bl/6J mice [32]. For 6 months, the mice were provided a diet that included 10% corn oil as a control, which could be supplemented with 0.5% CLA, 5% FO, or a mix of 0.5% CLA and 5% FO. Mice administered with CLA showed better BMD in multiple regions, along with decreased body weight, increased hind limb lean mass (HLLM), and decreased body fat mass (BFM). FO-fed mice, on the other hand, displayed increased insulin sensitivity and BMD without appreciable changes in BFM or HLLM. In aging mice, the combination of CLA and FO administration produced the best results, lowering inflammation and oxidative stress and enhancing BMD, HLLM, liver function, and bone marrow adiposity while also boosting body weight, BFM, and insulin sensitivity.

Further exploring the effects of CLA isomers, Rahman et al. (2011) investigated their impact on age-associated bone loss in 12-month-old female C57BL/6 mice. Over 6 months, mice were fed diets containing 10% corn oil (CO), 0.5% c9t11-CLA, 0.5% t10c12-CLA, or a mix of both isomers. The t10c12-CLA diet significantly increased bone mineral density (BMD) in the femur, tibia, and lumbar spine compared to CO or c9t11-CLA diets. This improvement was linked to reduced osteoclastogenic factors (RANKL, TRAP5b, TNF-α, and IL-6), lower bone marrow adiposity, and suppressed osteoclast differentiation. These findings highlight the t10c12-CLA isomer as particularly effective in mitigating bone loss by targeting both osteoclast activity and bone marrow adiposity [33].

An independent study investigated the effects of CLA on bone formation and pre-adipocyte differentiation in an osteoporosis-induced mouse model. In this study, a CLA mix was administered at a dose of 0.8 g/kg/day for 2 months via intragastric delivery. The control group received saline, and there was also an untreated positive control group. After the treatment period, bone health was evaluated through micro-computed tomography (micro-CT) and histological analyses. Results showed that CLA supplementation led to an increase in bone mineral density (BMD) and trabecular thickness (Tb.Th), indicating improved bone structure. Additionally, CLA reduced the number of adipocytes in the bone marrow, suggesting it helped mitigate bone marrow adiposity, a condition linked to bone degeneration [5]. These findings highlight the beneficial effects of CLA on bone structure and marrow adiposity. Building on this, Chaplin et al. (2015) explored the combined effects of CLA and calcium on bone health and energy metabolism in C57BL/6J mice. In their study conducted in 2015, C57BL/6J mice were used to assess the effects of CLA and calcium (Ca) on both bone health and energy metabolism. The mice were divided into five groups: control, high-fat diet (HF), HF + CLA, HF + Ca, and HF with both compounds (CLA + Ca). After 56 days of treatment, it was found that CLA alone was associated with decreased tibia weight and had minimal impact on bone markers. However, calcium, either on its own or combined with CLA, helped maintain bone weight and promoted the expression of key bone formation genes, including Bglap2 and Col1a1. Additionally, calcium influenced energy metabolism by affecting leptin and adiponectin receptors in bone tissue. These results suggest that calcium supplementation, especially when combined with CLA, supports both weight loss and bone metabolism [34].

Emerging evidence suggests that the effects of CLA on bone health may vary depending on its isomeric form. While several studies have demonstrated the overall benefits of CLA in improving bone structure and reducing marrow adiposity, the specific roles of individual CLA isomers in osteoblast activity remain an area of interest. In an in vitro study, SaOS-2 cells, which are human bone-derived osteosarcoma cells, were used to assess the effects of individual and mixed CLA isomers on bone formation. The CLA was introduced in the form of two isomers: 9cis, 11trans and 10trans, 12cis CLA. Results showed that the 9cis, 11trans CLA isomer significantly increased both the number and size of mineralized bone nodules—markers of bone formation—while the 10trans,12cis isomer did not show the same effect. Additionally, alkaline phosphatase (ALP) activity, an indicator of early osteoblast differentiation, increased variably with the 9cis,11trans CLA isomer. These findings highlight isomer-specific effects of CLA on bone health, particularly in promoting bone formation [35].

Murine mesenchymal stem cells were used to explore the effects of CLA on bone marrow adiposity and bone formation in another study. The CLA was supplied in two isomeric forms: trans-10, cis-12 CLA and cis-9, trans-11 CLA. The trans-10, cis-12 CLA isomer significantly inhibited adipogenesis and promoted osteoblastogenesis from mesenchymal stem cells, while the cis-9, trans-11 isomer had no such effect. The trans-10, cis-12 CLA’s inhibition of adipogenesis was mediated by PPARγ, a key regulator of fat cell development, but its promotion of osteoblast differentiation was independent of PPARγ. Additionally, this CLA isomer positively affected the osteoclastogenesis inhibitory factor, potentially influencing the bone-resorption processes [13]. The positive impact of CLA on bone health across various pre-clinical models is summarized in Table 1.

### 3.2. Clinical Trial

Conjugated linoleic acid (CLA) has demonstrated promising effects on calcium and bone metabolism in pre-clinical studies, including animal models and cell culture experiments. These findings suggest that CLA may play a role in supporting bone health by modulating bone formation and resorption. However, the translation of these benefits to human studies remains inconsistent. While some clinical trials indicate potential positive effects, others report neutral or even negligible outcomes [17]. Clinical studies have assessed the impact of commercially available CLA supplements, which typically consist of a roughly 50:50 blend of 9- and 10-CLA isomers and dairy products naturally enriched with CLA. In an 8-week intervention study involving 60 healthy men, supplementation with a 3.0 g CLA isomer blend (50:50% cis-9,trans-11:trans-10,cis-12 isomers) did not influence biomarkers of bone formation, such as serum osteocalcin and bone-specific alkaline phosphatase, nor did it affect biomarkers of bone resorption, including serum C-telopeptide, urinary *N*-telopeptide, urinary pyridinoline, and urinary deoxypyridinoline. Additionally, there were no changes in serum calcium concentrations or urinary calcium concentrations compared to the placebo [16]. While these findings suggest that CLA may not exert a significant impact on bone metabolism in healthy men, this lack of effect may be attributed to the relatively short study duration, insufficient CLA exposure, or the absence of underlying inflammatory conditions and elevated bone resorption within the study population. Researchers have explored CLA’s effects in populations at higher risk of bone loss, such as postmenopausal women. Postmenopausal women often experience significant bone loss, especially in the 1st decade after menopause, due to declining estrogen levels and increased inflammation. This leads to higher osteoclast activity and bone resorption, raising the risk of osteoporosis [37]. To explore whether conjugated linoleic acid (CLA) could mitigate these effects, a double-blind, placebo-controlled trial was conducted with 76 healthy postmenopausal women. Participants received 3.2 g/day of CLA isomer blend for 12 weeks, but the results showed no significant changes in bone formation, resorption markers, PTH levels, urinary calcium, creatinine, or IL-6 levels [38].

Several cell culture and animal studies indicate that CLA improves bone markers, but limited research exists on CLA’s impact on bone markers in autoimmune diseases, particularly rheumatoid arthritis, in humans. Aryaeian et al. conducted a study on the impact of CLA on bone markers in rheumatoid arthritis, a widespread autoimmune disease known for its chronic nature and heightened risk of osteoporosis [39]. The study confirmed that CLA benefits bone markers in rheumatoid arthritis patients, indicating its potential to prevent osteoporosis in this population. Furthermore, the same research group found that the combination of CLA and Vitamin E significantly improved anti-inflammatory outcomes in active rheumatoid arthritis, further supporting CLA’s role in reducing inflammation-related bone loss [40].

Beyond its potential effects on bone health, CLA supplementation has also been investigated in combination with other nutrients to enhance overall musculoskeletal outcomes. For instance, CLA has been explored alongside creatine monohydrate, a popular supplement known for improving exercise performance. A study by Pinkoski et al. found that co-supplementing CLA with creatine during resistance training increased lean tissue mass and reduced muscle protein breakdown compared to a placebo [41]. However, a study by Tarnopolsky et al. found no beneficial effects on bone turnover markers or BMD with creatine plus CLA supplementation or resistance training [42].

In contrast, CLA supplementation has shown some potential in reducing body fat, which could indirectly benefit bone health. Since obesity is known to negatively affect bone health, evidence points to a complex and potentially causal link between obesity and bone health [43]. Adipose tissue and bone are closely connected, as both fat and bone cells originate from the same stem cells within the bone marrow [44]. Visceral adipose tissue (VAT), in particular, is negatively associated with BMD, suggesting that high VAT may reduce BMD and increase the risk of bone-related diseases [45]. A double-blind, randomized, placebo-controlled trial by Mądry et al. involving obese women demonstrated that 12 weeks of CLA 3 g of CLA (50:50 cis-9, trans-11, and trans-10, cis 12 isomers) supplementation resulted in decreased levels of visceral, android, and gynoid adipose tissues, potentially mitigating VAT’s detrimental effects on BMD [46]. Excess body fat, particularly visceral fat, is known to negatively influence bone health by promoting inflammation and altering bone metabolism. In a study conducted by Chang et al., daily supplementation with 3.2 g of CLA in Chinese adults with high body fat was found to be effective in maintaining muscle mass, particularly in the trunk region [47]. These findings offer new insights, suggesting that CLA supplementation may play a role in preserving trunk muscle mass and supporting better bone mineral density. A higher muscle mass in the trunk area has been linked to improved spinal bone mineral density [48] and a reduced risk of kyphosis [49]. While several studies have reported positive effects of CLA on bone health, not all findings are consistent. However, a parallel randomized controlled trial was conducted to evaluate the impact of CLA supplementation on bone mineral density (BMD) and bone mineral content (BMC) in overweight and obese women. The study found that a 3-month regimen of 3 g/day CLA supplementation did not result in any improvement in bone health [50]. Similarly, a 6-month clinical trial in healthy, overweight, and obese adults (CLA 3.4 g/day) found no significant effect on BMC despite increases in lean body mass [51].

Obesity, with its associated inflammation, can interfere with bone remodeling, causing imbalances in bone maintenance and increased bone loss. Additional factors like oxidative stress, insulin resistance, shifts in gut microbiota, and hormonal changes may also contribute [52]. Oxidative stress, commonly elevated in obesity, leads to a decline in the function of both enzymatic and non-enzymatic antioxidants, resulting in increased bone degradation [53]. However, a study by Kim et al. evaluated the effect of CLA supplementation on antioxidant metabolism in healthy overweight/obese Korean individuals and found that 8 weeks of CLA at a dose of 2.4 g/day did not produce significant changes in lipid peroxidation or antioxidant metabolism [54]. Another key factor influencing bone health is interleukin-6 (IL-6), a pro-inflammatory cytokine that stimulates osteoclast formation [55]. Elevated IL-6 levels are commonly associated with increased bone resorption, leading to conditions such as osteoporosis, especially in cases of estrogen deficiency [56] and severe skeletal diseases [57,58,59]. Clinical studies have shown that CLA can reduce IL-6 levels, thereby potentially mitigating the negative impact of inflammation on bone health. In a double-blind clinical trial involving patients with Chronic Obstructive Pulmonary Disease (COPD), CLA supplementation was shown to significantly decrease serum IL-6 levels, thus improving overall health [60].

Studies examining CLA supplementation’s effects on bone health in healthy individuals, beyond overweight or obese subjects, have also yielded mixed results. Brown et al. compared the effects of beef and dairy products from pasture-fed cattle, which are naturally high in CLA, to those from grain-fed cattle, which are low in CLA, over 56 days in healthy female participants. The study found no significant changes in bone mineral density (BMD) and bone mineral content (BMC) [61]. A cross-sectional study explored the relationship between c9, t11 CLA levels in red blood cells, hypothesizing that higher CLA levels would increase bone mass. According to the study, men with RBC c9, t11 CLA levels above the median had higher whole body bone mineral density (BMD) and a greater percentage of whole body lean mass (WBL) [62]. The findings are summarized in Table 2.

## 4. Enhancing CLA Efficacy Through Nanoparticle-Delivery Systems

CLA encounters several challenges that restrict its effectiveness as a bioactive compound. Key issues include low water solubility and inadequate absorption, which significantly hinder its bioavailability [24]. Furthermore, its application in the food industry is limited by these solubility concerns and its susceptibility to oxidation, making it difficult to utilize CLA effectively as a functional ingredient [64]. Additionally, some drugs, including CLA, struggle to penetrate cell membranes effectively, leading to inadequate concentrations at target sites. To compensate, higher doses are often needed, which can result in increased toxicity and numerous unwanted side effects [65,66]. Although CLA is primarily absorbed in the small intestine, its physiological activity often diminishes or is lost before reaching this site. Additionally, despite being viewed as a natural alternative to conventional medications with relatively mild side effects, CLA has been linked to digestive issues such as nausea, upset stomach, and loose stools, primarily due to its instability and poor absorption [67]. Developing an effective carrier system for CLA encapsulation is essential to enhance its solubility, stability, and bioactivity. However, targeted drug-delivery systems can deliver sufficient drug concentrations directly to specific tissues or cells, enhancing bioavailability while minimizing the side effects typically associated with high doses [68,69]. Nanoparticle-delivery systems provide an effective solution to the challenges related to the bioavailability and stability of bioactive compounds like CLA [26,69]. Nanoparticles offer versatile benefits across various fields, from enhancing drug delivery and targeted therapies in medicine to improving water purification, agricultural efficiency, and environmental sustainability through advanced filtration, catalysis, and nutrient-delivery systems [69,70,71,72,73,74,75]. Utilizing nano-carrier systems for the encapsulation of fatty acids is a powerful method to boost their bioactivity and ensure precise, controlled delivery.

Carrier systems such as liposomes, nanostructured lipid carriers (NLCs), nano-emulsification, and cyclodextrins have proven to be highly suitable for this purpose, as they protect CLA from degradation and improve its delivery. Figure 3 illustrates various nanocarrier systems utilized for the nanoencapsulation of CLA. Liposomes, as versatile nano-carriers, significantly influence the encapsulation efficiency and release kinetics of CLA through their structural properties and formulation parameters [76,77,78]. The spherical shell of liposomes encases an aqueous interior that can contain peptides, proteins, hormones, enzymes, antibiotics, antifungals, and anticancer agents, making them vital for drug delivery. Their composition, specifically the phospholipid bilayer, allows for the encapsulation of both hydrophilic and hydrophobic compounds, including CLA [79]. The size of liposomes and the number of bilayers affect drug encapsulation. Larger liposomes enhance encapsulation efficiency for hydrophilic compounds, while an increased number of bilayers tends to decrease it [80]. Injecting drug-loaded liposomes reduces the required dose and administration frequency, minimizing toxicity and enhancing biodistribution, pharmacokinetics, and pharmacodynamics by prolonging the drug’s presence in the bloodstream [81]. Liposomes made from unsaturated phosphatidylcholine are more permeable but less stable, while those from saturated phospholipids, like dipalmitoyl phosphatidylcholine, result in rigid, less permeable bilayers [82,83]. The surface charge of liposomes also plays a critical role in their interactions with biological membranes. Positively charged liposomes enhance cellular uptake and encapsulation efficiency due to electrostatic attraction, while negatively charged liposomes experience a reduced uptake from repulsion with cell membranes [84]. In a previous study, two liposomal formulations containing 17% and 33% CLA were developed using a safe, scalable method, demonstrating high stability with no degradation of CLA over 30 days at 4 °C, and showing potential for enhanced fatty acid digestibility and bioavailability [85]. Additionally, another study reported that, due to their structure, liposomes can increase the solubility of lipophilic molecules in the gastrointestinal tract, thereby enhancing their bioavailability [86]. This same study found that lyophilized liposomes provided a protective effect on CLA isomers, with encapsulation efficiency remaining consistent between fresh and lyophilized samples. Moreover, CLA-loaded liposomes exhibited higher ordering in the inner part of the bilayer, indicating a more tightly packed membrane, which may contribute to the increased stability observed [86].

Continuing with lipid-based carriers, nano-structured lipid carriers (NLCs), as the second generation of lipid nanoparticles, represent a significant advancement due to their nano-scale particle size [87]. Unlike solid lipid nanoparticles, NLCs utilize a blend of both liquid and solid lipids in their matrix. This unique composition allows for higher encapsulation efficiency (EE%) and loading capacity (LC%) for compounds like CLA while also enhancing drug release properties and improving stability [88]. NLCs were found to exhibit higher encapsulation efficiency compared to solid lipid nanoparticles (SLNs) while also maintaining a smaller particle size, further enhancing their effectiveness in delivering compounds like CLA [89]. A study explored the use of NLC as an effective nano-carrier system for delivering CLA, assessing its protection against oxidation and heat [87]. The findings revealed that encapsulating CLA in NLC significantly enhanced its stability in pasteurized low-fat milk, reducing oxidation and the formation of harmful secondary products like malondialdehydes. Overall, the NLC formulation improved CLA’s resilience to thermal processes and environmental conditions. Encapsulation in cyclodextrins (CDs) also presents significant advantages in improving CLA’s oxidative stability and bioavailability. Cyclodextrins (CDs) are natural cyclic oligosaccharides that enhance drug bioavailability by forming inclusion complexes with poorly water-soluble molecules through hydrophobic interactions within their cavities [90]. Their unique structure, characterized by a hydrophobic cavity and a hydrophilic exterior, plays a significant role in influencing encapsulation efficiency and release kinetics. The cavity size of CDs significantly influences their capacity to encapsulate hydrophobic drugs. Beta-cyclodextrin is effective for smaller hydrophobic drugs, while gamma-cyclodextrin can encapsulate larger ones [91,92]. By enclosing these drugs, cyclodextrins improve their solubility and stability, leading to better bioavailability. A study found that α-, β-, and γ-CDs effectively protect CLA from oxidation, with α-CD providing the best protection at a 1:4 mole ratio. β-CD and γ-CD demonstrated similar protective effects at a 1:6 mole ratio, with β-CD exhibiting slightly superior performance [93]. The protective capabilities of the CDs were confirmed by reduced peroxide values (POV) in the formulations. Another study used β-cyclodextrin (β-CD) as a carrier for encapsulating CLA, achieving a high encapsulation efficiency of 97.8% and a loading of 10.5%. Importantly, β-CD significantly improved the oxidative stability of CLA, with only 35% of CLA oxidized after 280 h, while free CLA underwent complete oxidation under the same conditions [93]. The findings are summarized in Table 3.

The choice of encapsulation technique significantly influences the physicochemical properties of CLA-loaded carriers. Figure 4 illustrates CLA nanoencapsulation techniques. For example, the microencapsulation of CLA using complex coacervation enhances the stability and sensory qualities of products like buttermilk by forming a protective coating around the CLA, which helps prevent oxidation and degradation [101]. Complex coacervation is a phase-separation technique involving the aggregation of oppositely charged biopolymers, such as gelatin (a cationic protein) and Arabic gum (an anionic polysaccharide), around oil droplets to form microcapsules. This process typically involves heating, mixing, and gradual cooling, which allows the biopolymers to interact and form a stable coating [102,103]. This method is considered effective for encapsulating omega oils, including microbial oil-CLA, in food products [104].

Another technique explored in drug delivery is emulsification, which allows for the encapsulation of both lipophilic and hydrophilic drugs or imaging agents in either the oil or aqueous phase. CLA, being hydrophobic, is dissolved in the oil phase of the nano-emulsion alongside stabilizing agents like surfactants or co-surfactants. Nano-emulsification creates a nonequilibrium colloidal system where the oil phase is dispersed into fine droplets, typically ranging from 20 to 500 nm [105,106]. This technique effectively formulates hydrophobic active molecules, enhancing their bioavailability. It has been widely applied to improve the solubility and delivery of various bioactive compounds, including lipids [106]. These nano-emulsions are well-suited for pharmaceutical applications as drug-delivery systems due to their potential to enhance bioavailability and reduce side effects, making them a promising approach for improving the therapeutic efficacy of CLA [107]. Nano-emulsions can be prepared using high-energy and low-energy methods. High-energy methods apply significant mechanical force to break large droplets into nanoscale sizes, creating emulsions with high kinetic energy [108]. These methods mainly include high-pressure homogenization and ultrasonication. Alternatively, Ultra-High Pressure Homogenization (UHPH) has emerged as a promising technique for creating stable submicron emulsions, enhancing the antioxidative stability of encapsulated compounds like CLA [65]. Ultrasonication stands out among high-energy methods for its operational simplicity and ease of cleaning [109,110]. It uses cavitation forces generated by ultrasonic waves to convert macroemulsions into nano-emulsions. By adjusting the energy input and duration via ultra-sonicators, precise control over particle size and emulsion stability is achieved [111]. A study explored the enhancement of oil-in-water emulsions for delivering CLA in functional foods using UHPH technology [66]. Emulsions were created at 200 MPa with UHPH and at 15 MPa using conventional homogenization (CH). The results indicated that UHPH emulsions had a smaller particle size and a more homogeneous microstructure compared to those made with CH. Additionally, UHPH produced a sterile emulsion with improved physical stability during storage, effectively maintaining optimal oxidative stability for up to 3 months. In parallel, low-energy emulsification methods are highly energy-efficient, utilizing the system’s internal chemical energy with minimal stirring to produce nano-emulsions effectively [112].Techniques like phase inversion and self-emulsification enable nano-emulsion formation. In self-emulsification, surfactant and co-solvent molecules diffuse from the dispersed to the continuous phase, creating turbulence and nano-sized droplets without altering the surfactant’s curvature [113]. In the phase inversion method, changes in the spontaneous curvature of surfactants, driven by parameters like temperature and composition, induce phase transitions [114]. Examples include Phase-Inversion Temperature (PIT), Phase-Inversion Composition (PIC), and the Emulsion-Inversion Point (EIP), which optimize surfactant behavior for efficient nano-emulsion production [111].

A study by Jie Yang et al. focused on developing starch-based nanoparticles using an emulsification method to protect CLA [98]. They utilized octenyl succinic anhydride (OSA)-modified starch and xanthan gum (XG) as a complex carrier for CLA in their in vivo experiments. This approach aimed to enhance CLA’s absorption in the intestinal tract while minimizing its loss in the stomach. The results demonstrated that the emulsion nano-particles achieved an impressive encapsulation efficiency of over 97% for CLA. Furthermore, the study revealed that these CLA-loaded nanoparticles exhibited minimal release in the stomach, with the majority of the CLA being released after entering the small intestine, indicating an effective sustained release performance. A previous study demonstrates that employing nano-emulsification with soybean lecithin as a surfactant enhances both the thermal stability and bioavailability of CLA [97]. This is further supported by research evaluating the anti-obesity effects of nano-emulsified water-soluble conjugated linoleic acid (*N*-CLA), which also shows improved bioavailability in water, highlighting the effectiveness of nano-emulsification in optimizing CLA for various applications [94]. This study showed that nano-emulsified CLA (*N*-CLA) outperformed regular CLA in reducing fat accumulation, likely owing to its enhanced absorption due to the fine emulsification process. These outcomes suggest that *N*-CLA offers a viable solution to improve the bioavailability of CLA, which is otherwise unstable in aqueous environments [94].

Adipocyte-targeted therapies are crucial for reducing local adipose tissue, yet the clinical efficacy of CLA as an anti-obesity agent in humans is less convincing than in animal studies [115]. This is mainly due to insufficient adipocyte targeting [116]. Nano-formulations have emerged as a promising solution to enhance the bioactivity of fatty acids through controlled delivery [87]. In a study by Hsu et al., CLA-loaded tocol nano-structured lipid carriers (NLCs) were designed to improve the anti-adipogenic properties of CLA [100]. The localized administration of these NLCs in obese rats led to significant reductions in body weight, total cholesterol, and liver-damage markers. The prepared NLCs were largely internalized in adipocytes, increasing CLA delivery by 5.5-fold. This enhanced uptake resulted in a more profound inhibition of adipocyte differentiation compared to free CLA, highlighting the superior efficacy of tocol NLCs in targeting adipose tissue and improving the anti-adipogenic effects of CLA.

Choosing the right emulsifiers is crucial for stabilizing nano-emulsions. Encapsulating CLA in a Pickering emulsion system offers practical benefits, as protein nanofibrils (PNFs) serve as effective stabilizers. Protein nanofibrils (PNFs) are distinctive self-assembled structures characterized by their fibrous morphology, high aspect ratio, flexibility, biodegradability, and biocompatibility [117]. These properties enable PNFs to serve as effective emulsifiers. Beyond conventional molecular emulsifiers, solid particles can also function as stabilizers in emulsions, resulting in what are known as “Pickering emulsions” [118]. These emulsions are valued for their cost-effectiveness, non-toxicity, and environmentally friendly attributes. These emulsions exhibit excellent stability across various pH levels, ionic strengths, and temperatures [119]. Additionally, PNFs enhance controlled drug delivery by enabling sustained release and targeted nutrient distribution [120]. Most methods for preparing PNFs typically involve heating acidified protein solutions, usually at pH 2.0, to temperatures exceeding 80 °C for prolonged periods [121]. A study explored the potential of whey protein isolate nanofibrils (WPNFs), created using the Microwave-assisted self-assembly (MASA) method, to form Pickering emulsions for the encapsulation of CLA [99]. The results indicated that the WPNFs/CLA Pickering emulsions maintained greater stability in terms of droplet size and zeta potential across a wide range of ionic strengths and temperatures. This demonstrates that emulsions stabilized by WPNFs are particularly effective as carriers for CLA, enhancing its solubility and improving its applications in biological and food contexts. In another study by Wenjian Cheng et al., biopolymer-coated oil droplets were developed as natural delivery systems for incorporating CLA into functional food and beverage products [96]. CLA emulsions were stabilized by mixed biopolymer coatings by dispersing caseinate-coated droplets in pectin solutions at pH 5.0, enhancing electrostatic attraction. The stability of the emulsions depended on the pectin-to-caseinate ratio; higher ratios improved aggregation and creaming stability due to increased steric and electrostatic repulsion. Gum Arabic is another natural stabilizer used for oil-in-water emulsions. In a study, three types of gum Arabic—one conventional (GA) and two matured variants (EM2 and EM10)—were evaluated for their performance at the CLA-water interface [95]. Among them, EM10 demonstrated the best emulsifying activity and provided the highest emulsion stability, even with a lower surface load.

To minimize degradation and maintain the bioactivity of CLA in nano-emulsions, several strategies can be employed. Incorporating natural antioxidants, like Vitamin E or ascorbic acid, helps prevent oxidation, while selecting appropriate emulsifiers provides robust stability [122]. Protective polymers, such as chitosan or alginate, can form a barrier against environmental stress, and optimizing droplet size through processing techniques reduces the surface area exposed to degradation [123,124]. Additionally, adjusting the pH and ionic strength can strengthen the emulsifier layer, enhancing the emulsion’s overall stability over time [70].

In summary, nanoparticle-delivery systems offer a promising approach to enhance the stability, bioavailability, and resistance to oxidation of CLA. By improving targeted delivery and reducing the need for high doses, these systems can mitigate side effects and optimize the therapeutic potential of CLA. As research advances, nanoparticle formulations are expected to play a crucial role in overcoming the limitations of CLA, paving the way for more effective applications across various fields.

## 5. Electrical Stimulation (ES) as an Effective Strategy in Maintaining Bone Health

Electrical stimulation has emerged as a promising strategy for maintaining and enhancing bone health, particularly in cases of osteoporosis, fracture healing, and bone regeneration [125,126]. It leverages the principle that bone is a piezoelectric material, meaning it generates electrical charges in response to mechanical stress, which plays a crucial role in bone remodeling [127,128]. Activation of Piezo1 initiates intracellular calcium signaling responsible for the activation of transcription factors NFAT and Yap1, as well as β-catenin, all of which are crucial for osteoblast differentiation, as well as the formation of bones [127,128]. Electrical stimulation impacts the process of bone regeneration by activating mechanosensitive channels such as Piezo1, which are critical for the process of bone regeneration due to their response to mechanical pressure fluctuations [129,130]. This approach has gained attention due to its non-invasive or minimally invasive nature and its potential to modulate cellular activity and promote osteogenesis. Electrical stimulation has been shown to upregulate osteoblast proliferation, differentiation, and mineralization, leading to increased bone formation. It promotes the expression of bone-related genes such as RUNX2, ALP, and COL1A1, which are critical for osteogenesis [131,132,133]. Some studies suggest that electrical stimulation can reduce osteoclast differentiation and activity, leading to decreased bone resorption [134,135]. This effect helps in maintaining bone mass and counteracting osteoporosis-related bone loss. Mesenchymal Stem Cells (MSCs) exposed to electrical stimulation tend to differentiate more towards the osteogenic lineage rather than adipogenic lineage, which is beneficial for maintaining bone density [136,137,138]. Bone health is closely linked to vascularization. Electrical stimulation has been found to promote angiogenesis by upregulating VEGF expression, which supports bone healing and regeneration [131,139]. Various types of electrical stimulation can be used in maintaining bone health [140,141]. In Direct Electrical Stimulation (DES), electrodes are directly implanted near the bone to deliver low-voltage electrical signals, primarily used in fracture healing. Pulsed Electromagnetic Fields (PEMFs) are a non-invasive technique that generates electromagnetic fields to stimulate bone formation. This has been widely used in clinical settings for treating non-union fractures and osteoporosis. In Capacitive Coupling (CC) and Inductive Coupling (IC), external electrodes or coils are used to induce an electric field within the bone, promoting bone cell activity without direct contact. In Functional Electrical Stimulation (FES), instead of bones, muscles around the bones are stimulated, indirectly enhancing bone loading and preventing disuse osteoporosis in patients with paralysis. This strategy has been often used in rehabilitation.

Bone health is a concerning factor as one ages, as it has some impact on our bone-reforming system and its speed. Osteoporosis is a commonly known issue where bones become weak and brittle, making them more susceptible to fractures. Along with medication and other supplements, electrical stimulation has a greater impact in fighting this common problem. Electrical stimulation can show its multifunctional role in improving critical cell processes responsible for bone remodeling, recovery, and regeneration [125,142,143]. It allows osteogenic differentiation in cells like MSCs and osteoblasts through the activation of the pro-osteogenic pathway, which, again, is helpful for mineralization and bone tissue formation. Investigations underline electrical stimulation as capable of inducing cellular alignment and elongation, which are relevant features for efficient osteogenic differentiation and interaction with scaffold materials [142,143]. Moreover, electrical stimulation enhances cell migration, which increases the electrotaxis, cell ingrowth into the scaffolds, and interaction with the host tissues. Moreover, cell adhesion and attachment to biomaterials have been improved, thus offering an environment that is more propitious for differentiation and metabolic activity. These characteristics emphasize the potential of electrical stimulation to enhance the efficiency and effectiveness of bone-healing and regenerative therapies [27]. In a recent study, piezoelectric BaTiO₃/Ti₆Al₄V (BT/Ti) scaffold exhibited enhanced bone regeneration by modulating the immune microenvironment [144]. The poled BT/Ti scaffold displayed good biocompatibility and promoted M2 macrophage polarization, which supported osteogenesis in vitro and in vivo. Under low-intensity pulsed ultrasound (LIPUS) stimulation, the scaffold further increased the presence of M2 macrophages and improved bone repair in a sheep model. Mechanistically, the scaffold inhibited pro-inflammatory MAPK/JNK signaling and activated oxidative phosphorylation and ATP production in macrophages. These findings reveal a novel immunoregulatory mechanism by which piezoelectric materials promote bone regeneration [139]. In a meta-analysis of 15 randomized sham-controlled trials, it was found that electrical stimulation significantly reduces pain and radiographic nonunion in bone healing, though it shows no clear benefit for functional improvement. The evidence supports its use as an adjunct therapy [145]. A pilot study evaluated a novel in situ electrical stimulation device for enhancing implant osseointegration in a rabbit model. Using alternating electric fields (150 mV, 20 Hz, 3×/day), the device significantly improved bone-implant contact, as shown by both 2D histomorphometry and 3D µCT analysis. The system allowed real-time monitoring and parameter adjustment, demonstrating its potential for future optimization [146]. While results showed enhanced osseointegration, further studies are needed to establish optimal stimulation protocols and dose-response relationships. It was also demonstrated that long-term, low-intensity stimulation supports osteogenesis and aids bone-defect repair [147].

### 5.1. Combination of Electrical Stimulation and CLA on Bone Health

Our findings demonstrate that CLA supports bone health by modulating osteoblastogenesis, osteoclastogenesis, and adipogenesis within the bone marrow microenvironment [12,17,29,30,32,33,148,149]. Specifically, the t10c12 isomer of CLA enhances osteoblast proliferation and differentiation by upregulating key osteogenic markers such as RUNX2, ALP, and osteocalcin while also promoting bone matrix mineralization, calcium deposition, and bone strength. Additionally, CLA inhibits osteoclast-mediated bone resorption by suppressing pro-inflammatory cytokines (e.g., TNF-α, IL-6, and RANKL) and downregulating NF-κB signaling. Since excessive bone marrow adiposity is linked to osteoporosis, CLA plays a crucial role in modulating lipid metabolism, shifting mesenchymal stem cell (MSC) differentiation toward osteoblasts rather than adipocytes, thereby enhancing bone formation. The t10c12 isomer has been particularly effective in reducing bone marrow fat accumulation. Since CLA and electrical stimulation (ES) share similar mechanisms in maintaining bone health, we propose integrating ES with CLA supplementation as a multimodal approach for bone regeneration, particularly for individuals at risk of osteoporosis (Table 4). This combination capitalizes on ES’s regenerative effects alongside CLA’s biological benefits, fostering enhanced bone formation and remodeling [126,145,146] (Figure 5).

Techniques such as Direct Current (DC) and Pulsed Electromagnetic Fields (PEMF) are two distinct forms of electrical stimulation that activate osteoblasts and accelerate new bone formation by replicating the intrinsic piezoelectric properties of bone tissue. The combined effect of electrical stimulation (ES) and CLA may optimize bone regeneration, leading to incremental increases in bone mineral density (BMD). This synergy is particularly beneficial in critical regions like the lumbar spine, where ES stimulates bone formation, while CLA further enhances bone formation and inhibits resorption [46].

### 5.2. Optimizing Osteogenesis and Diminishing Inflammation with Combined Electrical Stimulation and Conjugated Linoleic Acid Techniques

The combined metabolic effects of CLA and electrical stimulation present a promising strategy for enhancing osteoblast activity and promoting osteogenesis. This electro-medical approach is particularly intriguing as it not only mitigates osteoclast-induced bone degeneration but also helps preserve mechanical bone strength. Electrical stimulation, in particular, supports the proliferation of osteoblasts, reinforcing weight-bearing regions such as the spine. By integrating these therapies, a comprehensive method for improving bone health will be achieved, addressing both bone formation and deterioration [142,143].

Additionally, it enhances the benefits of CLA by applying it to areas where its effects are less apparent while also creating a cellular environment that supports bone regeneration. This combined approach can yield positive outcomes for osteoporotic patients or those recovering from osteoporosis, promoting bone stability and structure more effectively than any single treatment. By addressing both bone formation and resorption, this method offers a comprehensive improvement in bone health. Furthermore, electrical stimulation (ES) can enhance the local effects of CLA by fostering a cellular environment that supports bone growth, particularly in areas where CLA’s impact is limited. When used together, these therapies should significantly boost therapeutic results in osteoporotic patients or those healing from fractures, helping to preserve bone density and integrity.

### 5.3. The Potential for Electrical Stimulation to Enhance Bone Tissue Absorption and CLA Effectiveness

Electrical stimulation influences cellular behavior, supports bone remodeling, and promotes osteogenesis, all crucial for bone regeneration. Electret materials, such as SiO_2_/PDMS composite membranes, are used to generate stable, long-lasting electrical fields that mimic the natural conditions of bone [150]. These membranes have been shown to enhance osteogenic differentiation and mesenchymal stem cell activity in both in vitro and in vivo models. Since continuous electrical stimulation accelerates bone regeneration, these materials hold promise for improving bone repair and regeneration in clinical settings [150]. Membranes containing electret material with a persistent electric charge can provide steady, long-term electrical stimulation. Medical devices for bone regeneration utilize electrode-incorporated membranes to deliver consistent electrical signals, promoting osteogenic differentiation and aiding in bone healing.

Electrical stimulation plays a crucial role in bone remodeling and regeneration by modulating cellular processes, including bone metabolism and osteoblast differentiation [125,126]. Both endogenous and externally applied electrical fields influence cellular-signaling pathways, such as extracellular signal-regulated kinase (ERK) and protein kinase A (PKA), which regulate osteoblast activity [151]. These mechanisms support osteoblast development, proliferation, and function, which are essential for bone repair and regeneration. Additionally, electrical currents, including alternating electric fields, have been shown to enhance the expression of osteogenic genes such as collagen type I and alkaline phosphatase, thereby promoting osteoblast activity [152,153,154]. Low-to-moderate voltage electric fields further facilitate bone growth and repair by increasing osteogenic gene expression and protein accumulation. Moreover, mechanical stress combined with electrical stimulation can trigger the release of growth factors, such as bone morphogenetic proteins (BMPs), further enhancing bone regeneration and remodeling [155,156,157].

In our studies, we have demonstrated that CLA enhances bone cell activity by promoting osteoblastogenesis and inhibiting osteoclastogenesis, leading to reduced inflammation and bone resorption [14,33]. Electrical stimulation works by modulating RANKL signaling, which is essential for bone resorption, thereby decreasing osteoclast relaxation time and stimulating osteoblast activity [35,140,158]. Pulsed electromagnetic fields (PEMF) have been shown to enhance bone healing by increasing cell activity and promoting blood flow to targeted bone areas. These methods can augment the reparative and anabolic activities of bone tissue, leading to improved bone regeneration [159,160].

Electrical stimulation has been shown to promote osteoblast activity while inhibiting osteoclast activity, thereby supporting bone growth and remodeling. Additionally, this mechanism aids in bone tissue regeneration and healing by facilitating the development of vascular networks. CLA also supports bone growth and remodeling by stimulating bone formation and inhibiting bone resorption. Both ES and CLA share common mechanisms that promote bone regeneration (Table 4). Based on this shared potential, we propose that the combined application of electrical stimulation and CLA may offer synergistic benefits in improving bone health in osteoporotic conditions. Nevertheless, further pre-clinical and clinical investigations are essential to validate the efficacy of this combinatorial approach.

## 6. Conclusions

In conclusion, CLA demonstrates significant potential as a therapeutic agent for enhancing bone health by influencing bone turnover, mineralization, and mechanical strength. Through its modulation of osteoblast activity (bone formation) and osteoclast activity (bone resorption), CLA contributes to balanced bone remodeling, which can improve bone density and quality. One of CLA’s notable effects is its ability to reduce visceral adipose tissue (VAT), thereby indirectly supporting bone health by reducing inflammation and fostering a healthier environment for bone cells.

While preclinical studies emphasize CLA’s promising effects, clinical trials in humans have shown mixed results. Some studies confirm that CLA benefits bone health by reducing inflammation and lowering IL-6 levels, which helps mitigate bone resorption and inflammation-related bone loss in conditions like rheumatoid arthritis and COPD. Additionally, CLA improves bone health in obese populations by reducing visceral adipose tissue (VAT), which, in turn, benefits bone health. However, other studies fail to report consistent improvements in bone mineral density (BMD), bone mineral content (BMC), or bone metabolism markers. These inconsistencies may arise from differences in dosages, isomer compositions, and study designs, as well as disparities between animal models and human studies. Moreover, CLA supplementation has been shown to maintain bone markers in healthy populations without significant improvements, which may be attributed to factors such as short study durations, limited CLA exposure, and the lack of inflammatory conditions and increased bone resorption in these populations.

Despite its potential, CLA faces challenges such as poor bioavailability caused by low solubility, stability issues, and oxidation. These limitations can be addressed through advanced delivery systems like liposomes, nano-structured lipid carriers (NLCs), solid lipid nanoparticles (SLNs), and cyclodextrins (CDs), which enhance CLA’s solubility, stability, and targeted delivery, thereby improving its therapeutic efficiency.

Combining conjugated linoleic acid (CLA) supplementation with electrical stimulation (ES) presents a promising strategy to maintain bone health by targeting multiple pathways involved in bone metabolism. We anticipate that when paired with electrical stimulation, CLA’s efficacy in promoting bone health will be further enhanced. ES techniques, such as Direct Current (DC) and Pulsed Electromagnetic Fields (PEMF) stimulate osteoblast activity, boost bone mineral density (BMD), and mitigate inflammation. CLA may prime osteoblasts for a better response to ES by modulating lipid metabolism and reducing oxidative stress. Both approaches may suppress osteoclast activity, preventing excessive bone breakdown. ES can increase ATP production, which may support the anabolic effects of CLA on bone tissue. Together, CLA supplementation and ES represent a synergistic approach to enhancing osteogenesis, reducing bone degradation, and accelerating bone repair.

This integrated strategy underscores the transformative potential of CLA, particularly when combined with advanced delivery systems and ES, as complementary interventions to promote bone health. These approaches are especially relevant for managing conditions like osteoporosis or inflammatory diseases that compromise bone strength. However, further research is crucial to refine CLA’s delivery systems, enhance its bioavailability, and establish effective therapeutic protocols to maximize its clinical potential in supporting bone health and recovery. More research is needed to determine the best CLA dosage and ES intensity/frequency for maximal benefits. While preclinical studies support the benefits, large-scale human trials are needed to confirm efficacy. The combination of CLA supplementation and electrical stimulation offers a promising, non-invasive strategy to support bone health by enhancing osteogenesis, reducing inflammation, and improving mechanotransduction. Further research could establish its clinical applications for osteoporosis prevention and bone-repair therapies.

## Figures and Tables

**Figure 1 nutrients-17-01395-f001:**
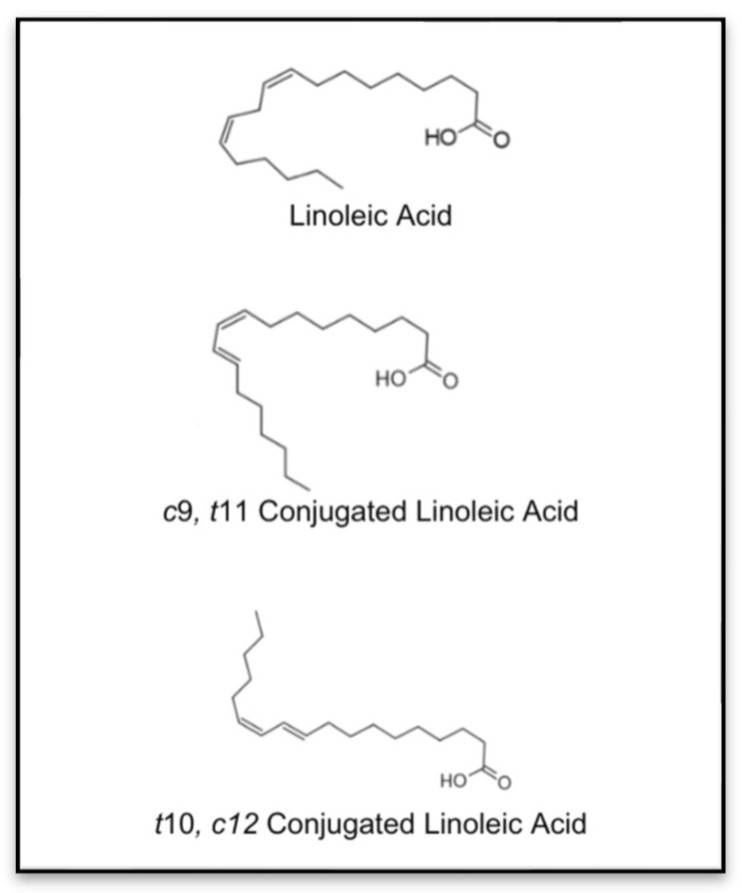
Chemical structures of linoleic acid and isomers of CLA [6].

**Figure 2 nutrients-17-01395-f002:**
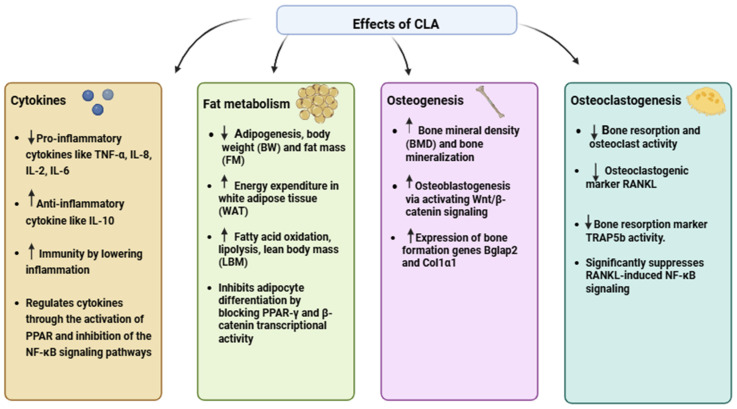
CLA’s effects on cytokine regulation, fat metabolism, bone formation, and bone resorption. (↑ indicates an increase; ↓ indicates a decrease).

**Figure 3 nutrients-17-01395-f003:**
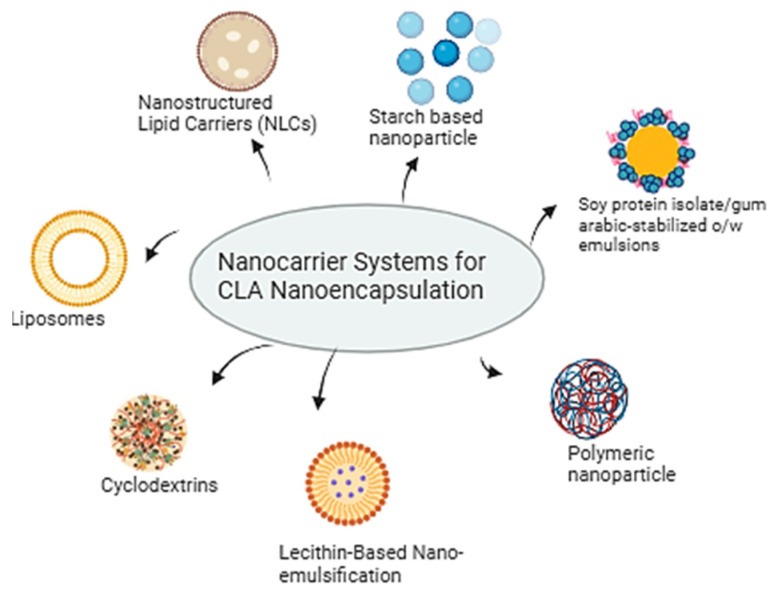
Overview of nano-carrier systems for CLA nano-encapsulation.

**Figure 4 nutrients-17-01395-f004:**
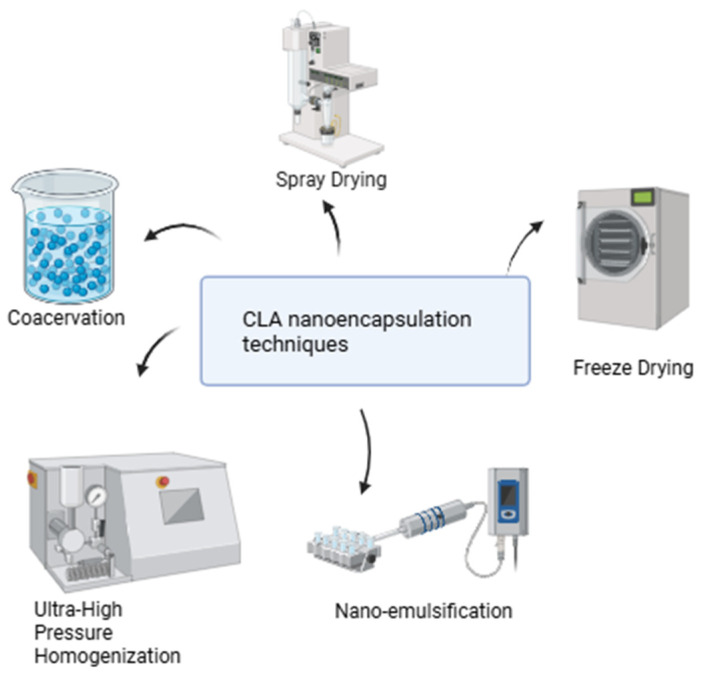
CLA nanoencapsulation techniques.

**Figure 5 nutrients-17-01395-f005:**
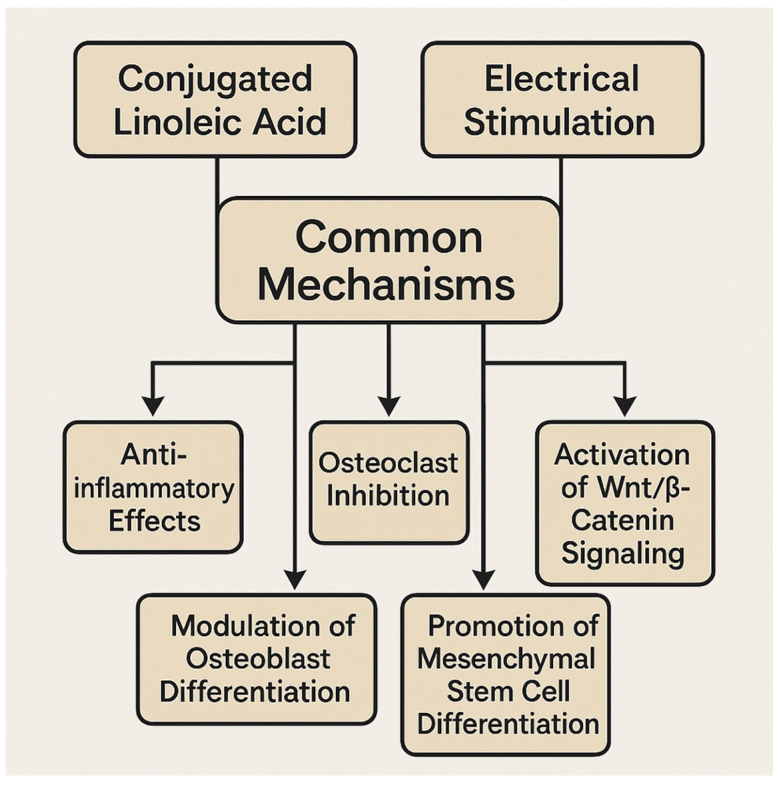
Shared mechanisms of CLA and ES on bone remodeling.

**Table 1 nutrients-17-01395-t001:** Beneficial Effects of CLA on Bone Health across Various Pre-Clinical Models.

Reference	Model Type	In Vitro/In Vivo	Treatment	Duration	Results
Kelly and Cashman, 2004 [31]	OVX rats	In Vivo	Basal AIN-93M diet with CLA (2.5, 5.0, or 10 g/kg). *n*- CLA replaced equivalent amounts of soybean oil (SBO). Fresh diets were prepared weekly and provided daily.	9 weeks	CLA decreased bone-resorption markers but had no impact on BMD. Higher doses were more effective.
I. Platt et al., 2007 [35]	Human SaOS-2 cells	In Vitro	CLA isomers (cis-9, trans-11 and trans-10, cis-12) and CLA mix at 25–100 μM. Bone nodules were stained using the von Kossa method. ALP activity was measured as an osteoblast differentiation marker.	Not specified	9cis,11trans CLA increased the number and size of mineralized bone nodules, whereas 10trans,12cis CLA had no such effect.
Rahman et al., 2007 [30]	14-month-old C57BL/6 mice	In Vivo	AIN-93M diet with: o 10% corn oil (CO) (control) o 9.5% CO + 0.5% CLA (50:50 mix of cis-9, trans-11, and trans-10, cis-12). Fresh diets were prepared weekly.	10 weeks	CLA increased BMD, reduced pro-inflammatory cytokines, decreased fat mass, and increased muscle mass.
Halade et al., 2011 [32]	12-month-old C57BL/6J mice	In Vivo	AIN-93G diet with: o 10% CO (control) o 5% fish oil (FO) + 5% CO o 0.5% CLA + 9.5% CO o 0.5% CLA + 4.5% CO + 5% FO. Diets were purged with nitrogen, stored frozen, and provided fresh daily.	20 weeks	CLA improved BMD and muscle mass but led to insulin resistance. CLA + FO combination reduced inflammation, improved BMD, and energy metabolism.
Rahman et al., 2011 [33]	12-month-old C57BL/6 mice	In Vivo	AIN-93M diet with: o 9.5% CO + 0.5% cis-9, trans-11 CLA o 9.5% CO + 0.5% trans-10, cis-12 CLA o 9.5% CO + 0.5% CLA mix (50:50 ratio). Fresh diets were prepared weekly and provided daily.	26 weeks	The t10,c12 CLA isomer significantly improved BMD and reduced osteoclastogenic factors and bone marrow adiposity.
Park et al., 2013 [9]	6-month-old ICR mice	In Vivo	Mixed CLA + Calcium (CLA: 80.7% purity; cis-9, trans-11 and trans-10, cis-12 isomers in 50:50 ratio). Provided ad libitum with AIN-93 diets. Fresh diets were prepared weekly and stored at −20 °C.	8 weeks	CLA increased BMD and bone strength in ovariectomized (OVX) mice. CLA + Calcium improved BMD and bone formation markers in comparison to controls
J. Kim et al., 2013 [13]	Murine mesenchymal stem cells	In Vitro	α-MEM medium supplemented with: o 0.5–50 μM trans-10, cis-12 CLA CLA mix.	4 weeks	The trans-10,cis-12 CLA isomer inhibited adipogenesis and promoted osteoblastogenesis, positively influencing bone-resorption processes.
Kim st al. 2014 [36]	n murinemesenchymal stem cells	In vitro	CLA isomers (cis-9, trans-11 and trans-10, cis-12) and CLA mix Osteoblastogenesis; adipogenesis; osteoclastogenesis	28 days	The trans-10,cis-12 CLA enhances osteoblastogenesis through a SMAD8-mediated mechanism; inhibiting adipogenesis independently of SMAD8; and reducing factors involved in osteoclastogenesis.
Rahman et al., 2014 [12]	8-week-old C57BL/6 mice	In Vivo	CLA diet (0.5% CLA; 50:50 mix of cis-9, trans-11 and trans-10, cis-12). Control: 0.5% safflower oil (SFO). Fresh diets were prepared weekly, stored at −20 °C, and offered ad libitum.	24 weeks	CLA prevented bone loss in OVX mice and stimulated new bone formation.
Chaplin et al., 2015 [34]	C57BL/6J mice	In Vivo	AIN-93 diet was supplemented with: o 10% CO (control) o 9.5% CO + 0.5% cis-9, trans-11 CLA o 9.5% CO + 0.5% trans-10, cis-12 CLA o 9.5% CO + 0.25% CLA mix. Diets were provided fresh daily to prevent rancidity.	8 weeks	CLA alone had minimal impact, but combined with calcium, it improved bone weight and expression of bone formation genes like Bglap2 and Col1a1.
Lin et al., 2017 [5]	C57BL/6J mice	In Vivo	Mixed CLA diet (concentrations not specified) supplemented with soybean oil. Freshly prepared diets were provided daily.	1 week	CLA increased BMD and reduced bone marrow adiposity.

**Table 2 nutrients-17-01395-t002:** Summary of outcomes of clinical studies published between 2007 and 2023 on the effect of CLA on bone health.

Reference	Study Population	Age/Mean ± SD Age	Treatment	Duration	Results
Tarnopolsky et al., 2007 [42]	39 community-dwelling, older adults	65–85 years	5 g creatine monohydrate (CrM) + 6 g of CLA (45% c9, t11; 45% t10, c12)	26 weeks	-The combination of CrM and CLA enhanced the benefits of resistance exercise in older adults, significantly improving strength (*p* < 0.001), functional capacity (*p* < 0.05), and muscular endurance compared to exercise alone.
Racine et al., 2010 [63]	53 overweight or obese children (BMI > 85th percentile)	6–10 years	Clarinol™ 3.0 g (80% CLA, 50% c9t11, 50% t10c12) (*n* = 28) Placebo: sunflower oil (*n* = 25)	30 weeks	-The CLA group experienced a reduction in total body bone mineral content (BMC) accrual, with bone mineral accretion lower in the CLA group compared to the placebo group. This indicates that CLA had no notable effect on improving bone formation.
Brown et al., 2011 [61]	18 healthy women BMI between 19–30	20–40 years	CLA diet: 1.17 g/d; control diet: 0.35 g/d	8 weeks	-No significant changes in bone mineral density (BMD) and BMC.
Deguire et al., 2012 [62]	54 community-dwelling adult men	19–53 years	1.5 g, 3.0 g cis-9, trans-11 CLA mixture Placebo: olive oil	17 weeks	-Men with higher RBC c9,t11 CLA levels showed increased BMD and lean mass
J. Kim et al., 2012 [54]	29 healthy overweight/obese Korean individuals	19–65 years	2.4 g/day CLA [36.9% of cis-9, trans-11 and 37.9% of trans-10, cis-12]	8 weeks	-No significant impact on lipid peroxidation or antioxidant metabolism was observed, indicating no effect on bone formation. No difference in plasma TRAP levels suggests no effect on bone resorption.
Darestani R et al., 2013 [38]	76 healthy postmenopausal women	45–65 years	CLA G80 containing 3.2 g isomer blend (50:50% cis-9, trans-11: trans-10, cis-12 isomers) Placebo: oleic sunflower oil	12 weeks	-CLA supplementation had no significant effects on markers of bone formation (serum osteocalcin, bone-specific alkaline phosphatase) or bone resorption (urine C-telopeptide). No changes were observed in PTH, urinary calcium, creatinine, or IL-6 levels, indicating no impact on overall bone metabolism or calcium.
Aryaeian et al., 2014 [40]	78 patients with active rheumatoid arthritis	18 and 69 years	1.25 g/day 80% CLA [2 g 50:50 mix of cis-9, trans-11 and trans-10, cis-12 glycerinated CLA]	13 weeks	-CLA’s anti-inflammatory effects help reduce inflammation-associated bone loss in rheumatoid arthritis (RA) patients, confirmed by reduced white blood cell (WBC) count, MMP-3, and TNF-α levels in active RA patients, indicating a beneficial impact on bone health.
Aryaeian et al., 2016 [39]	52 patients with active rheumatoid arthritis	19–69 years	2 g of 9-cis 11-trans isomer and 10-cis 12-trans isomer in ratio of 50 −50 CLA in glycerinated form	12 weeks	-CLA benefits bone health in rheumatoid arthritis patients by positively affecting telopeptide C and osteocalcin levels
Aslani et al., 2020 [60]	82 COPD patients	60–65 years	3.2 g of CLA	6 weeks	-CLA supplementation significantly reduced serum IL-6 levels, modified inflammatory markers, and improved overall health status, confirming a positive influence on bone health
Chang et al., 2020 [47]	66 Chinese adults with elevated body fat percentage	18–45 years	3.2 g/day CLA Placebo: Sunflower oil	12 weeks	-CLA supplementation helps preserve muscle mass, particularly in the trunk region, and contributes to better BMD.
Jamka et al., 2023 [50]	74 Caucasian Obese women	50–55 years	3 g (80% CLA, 50% of cis-9, trans-11, 50% of trans-10, cis-12 isomers) (*n* = 37) Placebo: sunflower oil (*n* = 37)	13 weeks	-The CLA group showed significant increases in BMC and BMD at the lumbar spine. There were no differences in BMC and BMD at the total body and femoral neck between the CLA and placebo groups.

**Table 3 nutrients-17-01395-t003:** Summary of studies from 2000 to 2024 using nanoparticle-encapsulated conjugated linoleic acid.

Reference	Nanoparticle Type	Loaded Compound	Findings
Kim, S. J et al., 2000 [93]	Alpha, beta, and gamma Cyclodextrins(CD)	CLA (48% cis-9, trans-11 and 48% trans-10, cis-12 isomers)	CLA/CD microencapsulation (1:4 mole ratio) fully protected CLA from oxidation, with α-, β-, and γ-CDs reducing peroxide values, confirming their protective role.
Kim, D et al., 2013 [94]	Nanoemulsified water-soluble conjugated linoleic acid (*N*-CLA)	CLA (>77% purity).	*N*-CLA outperformed CLA in reducing fat accumulation due to better absorption. It effectively lowered body weight, improved blood and liver lipid profiles, and enhanced CLA bioavailability, making it a potential anti-obesity agent.
Xiang, S et al., 2015 [95]	Gum Arabic (biopolymer emulsifier)	80% pure CLA	Among the tested gum Arabic types (conventional GA, matured EM2/EM10), EM10 showed superior emulsifying activity and stability despite a lower surface load.
Cheng, W et al., 2016 [96]	Biopolymer-coated oil droplets	75–80% CLA (50:50 ratio of c9,t11 and t10,c12 isomers)	Mixed biopolymer-coated emulsions demonstrated thermal stability (90 °C/20 min). Stability was pectin-to-caseinate ratio-dependent, with higher ratios improving aggregation and creaming resistance via enhanced steric/electrostatic repulsion.
Heo, W et al., 2016 [97]	Soybean lecithin	CLA (38.6% cis-9, trans-11; 43.3% trans-10, cis-12; 3.5% other isomers)	Nano-emulsification with soybean lecithin improved CLA’s thermal stability, bioavailability, and enhanced CLA absorption in the small intestine.
Vélez, M. A et al., 2017 [85]	Soy phosphatidylcholine (PC) liposomes	CLA isomers 9c, 11t and 10t, 12c	CLA-loaded nanoparticles remained stable for 30 days at 4 °C, preventing CLA degradation. They enhanced membrane fluidity, improved digestibility and bioavailability, and achieved over 80% encapsulation efficiency.
Vélez, M. A et al., 2019 [86]	Soy phosphatidylcholine (PC) liposomes	CLA isomers 9c, 11t and 10t, 12c	CLA-loaded liposomes showed improved stability and smaller size compared to control liposomes during storage while efficiently preserving CLA isomers with high encapsulation efficiency.
Hashemi, F.S et al., 2020 [87]	Nanostructured lipid carrier (NLC)	CLA with 80% purity, a mixture of 9-cis, 11-trans and 10-trans, 12-cis isomers.	Encapsulating CLA in NLC systems improved protection against oxidation, reduced secondary oxidation products like malondialdehydes, and enhanced stability against thermal processes, environmental conditions, and oxidation.
Yang, J et al., 2020 [98]	Octenyl succinic anhydride (OSA)-modified starch and xanthan gum (XG)	CLA (80% purity, cis-9, trans-11/cis-10, trans-12 octadecadienoic acids, and linoleic acid < 1%)	Encapsulation efficiencies exceeded 97%, ensuring effective CLA entrapment. Release studies showed minimal CLA release in the stomach, with most released in the small intestine.
Jiao, Q et al., 2021 [99]	Pickering Emulsion with Whey Protein Nanofibers	CLA	Whey protein nanofibrils (WPNFs) effectively encapsulated CLA, improving its water solubility and addressing key delivery challenges of lipophilic bioactives.
Hsu, C. Y et al., 2024 [100]	Tocol nanostructured lipid carriers (NLCs)	CLA (41.2% c9-t11 and t9-c11 CLA, 44.1% t10-c12 CLA, and 9.4% t10-c12)	CLA-loaded NLCs more effectively reduced fat storage, TG levels, and adipokine expression in adipocytes than free CLA.

**Table 4 nutrients-17-01395-t004:** Common Mechanisms in Bone Regeneration: CLA vs. Electrical Stimulation.

Mechanism	Conjugated Linoleic Acid (CLA)	Electrical Stimulation (ES)
Anti-inflammatory Action	Suppresses TNF-α, IL-6, and COX-2 expression; decreases osteoclast formation	Decreases pro-inflammatory cytokines (e.g., IL-1β, TNF-α); promotes a healing environment
Osteoblast Differentiation	Upregulates Runx2, ALP, osteocalcin; enhances osteogenesis	Activates MAPK/ERK and calcium signaling; promotes ALP activity and mineralization
Osteoclast Inhibition	Inhibits RANKL and upregulates OPG; reduces bone resorption	Alters cellular bioelectric signals to suppress osteoclast activity
Wnt/β-Catenin Pathway	Indirect activation; supports osteoblast lineage commitment	Direct activation; promotes β-catenin nuclear translocation and gene expression
MSC Differentiation	Stimulates MSCs to become osteoblasts over adipocytes	Guides MSC fate via physical/electrical cues toward osteogenesis
Oxidative Stress Regulation	Acts as an antioxidant; protects osteoblasts from ROS-induced apoptosis	Moderate ES reduces oxidative stress, enhancing osteoblast survival and function

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
