# Peer review of "Enhancing Bone Health with Conjugated Linoleic Acid: Mechanisms, Challenges, and Innovative Strategies"

_nutrients, 2025, doi:10.3390/nu17081395_

Round 1

Reviewer 1 Report

Comments and Suggestions for Authors

 This review explores the combined use of CLA  supplementation and electrical stimulation as a novel approach to improving bone health,  particularly in osteoporosis management. By integrating CLA's biological effects with the  regenerative potential of electrical stimulation, this multimodal strategy offers a promising method for enhancing bone restoration, with significant implications for clinical applications in bone health.

The introduction summarises the advantages and disadvantages of CLA on health, the problems that determine its use and possible solutions. The bibliography used is marked

The methodology used for this narrative review is not indicated.

The preclinical and clinical studies carried out that indicate the effects achieved on bone health are exhaustively indicated. Tables help to understand the results obtained

The second part of the review refers to the delivery systems of CLA to avoid its disadvantages as well as the combination of these techniques with electrical stimulation to improve its effects.

A graphical representation of the mechanisms of action of CLA on bone cells and the association with electrical stimulation would be useful.

Bibliographic citations should be adapted to the journal's standards.

Author Response

Comments and Suggestions for Authors

 This review explores the combined use of CLA  supplementation and electrical stimulation as a novel approach to improving bone health,  particularly in osteoporosis management. By integrating CLA's biological effects with the  regenerative potential of electrical stimulation, this multimodal strategy offers a promising method for enhancing bone restoration, with significant implications for clinical applications in bone health.

The introduction summarises the advantages and disadvantages of CLA on health, the problems that determine its use and possible solutions. The bibliography used is marked

The methodology used for this narrative review is not indicated.

Response: We searched for all published preclinical and clinical studies related to this topic available in PubMed and Science Direct till to date and included in our narrative review.

The preclinical and clinical studies carried out that indicate the effects achieved on bone health are exhaustively indicated. Tables help to understand the results obtained

The second part of the review refers to the delivery systems of CLA to avoid its disadvantages as well as the combination of these techniques with electrical stimulation to improve its effects.

  1. A graphical representation of the mechanisms of action of CLA on bone cells and the association with electrical stimulation would be useful.

Response. Thank you for your valuable suggestion. we have added Figure 2, which provides a graphical representation of the effects of CLA on bone, including its influence on bone formation, bone resorption, fat metabolism, and cytokine regulation. This figure has been included in the introductory section on page 3, starting from line 121, and has been highlighted in green for your convenience.

A separate graphical illustration on the possible association between CLA and electrical stimulation effect is now included as a Figure 5 and a Table 4 in our revised manuscript.

  1. Bibliographic citations should be adapted to the journal's standards.

Response: Thank you for your valuable feedback. We have made every effort to format the references according to the journal's guidelines. However, the exact citation style is not available in the reference management software we are using, and manually adjusting all references would be quite time-consuming. If the MDPI team could assist in converting the references to the exact required format, it would be greatly appreciated. In our previous submission with MDPI, we received similar support, which was extremely helpful.

All changes are highlighted in green for your convenience.

Reviewer 2 Report

Comments and Suggestions for Authors

Khandoker Hoque et al. did a nice job in reviewing the effects of conjugated linoleic acid (CLA)on bone health and summarizing the current strategies to overcome the low water solubility and poor bioavailability problems. I have some recommendations to the authors:

  1. Introduction: I recommend the authors to draw a graph to describe the roles of CLA on cytokines, fat metabolism, osteogenesis, and osteoclastogenesis. Especially it is better to show which signaling pathways that CLA regulates leading the biological effects.
  2. Enhancing CLA efficacy through nanoparticle delivery systems: I recommend the authors summarize previous nanoparticle studies as a Table 3, similar as Table 1&2.
  3. Electrical Stimulation as an effective strategy in maintaining bone health: I recommend the authors to draw a graph to show the effects of electric stimulation on bone health. Also, this section lacks some citation to previous studies. For example, from line 689 to 692, the authors list the effects of electric stimulation on ERK, PKA signaling without any citation. From line 716 to line 718, the authors mentioned that “Electric stimulation may influence CLA-related biochemical pathways by reducing pro-inflammatory cytokines”. The word “may” is vague, and this paragraph did not show any citation to demonstrate the effects of electric stimulation on cytokine expression in previous studies. Please cite previous publications and only draw conclusion based on previous studies.

Author Response

Comments and Suggestions for Authors

KhandokerHoque et al. did a nice job in reviewing the effects of conjugated linoleic acid (CLA)on bone health and summarizing the current strategies to overcome the low water solubility and poor bioavailability problems. I have some recommendations to the authors:

  1. Introduction: I recommend the authors to draw a graph to describe the roles of CLA on cytokines, fat metabolism, osteogenesis, and osteoclastogenesis. Especially it is better to show which signaling pathways that CLA regulates leading the biological effects.

Response. Thank you for your insightful suggestion. we have added Figure 2, which illustrates the role of CLA in regulating cytokines, fat metabolism, osteogenesis, and osteoclastogenesis, along with the associated pathways. This figure has been included in the introductory section on page 3, starting from line 121, and has been highlighted in green for your convenience.

  1. Enhancing CLA efficacy through nanoparticle delivery systems: I recommend the authors summarize previous nanoparticle studies as a Table 3, similar as Table 1&2.

Response: Thank you for your valuable suggestion. We have added Table 3, summarizing previous nanoparticle studies in a format similar to Tables 1 and 2, including all relevant details. This table has been included on page 16, starting from line 466, and has been highlighted in green for your convenience.

  1. Electrical Stimulation as an effective strategy in maintaining bone health: I recommend the authors to draw a graph to show the effects of electric stimulation on bone health. Also, this section lacks some citation to previous studies. For example, from line 689 to 692, the authors list the effects of electric stimulation on ERK, PKA signaling without any citation. From line 716 to line 718, the authors mentioned that “Electric stimulation may influence CLA-related biochemical pathways by reducing pro-inflammatory cytokines”. The word “may” is vague, and this paragraph did not show any citation to demonstrate the effects of electric stimulation on cytokine expression in previous studies. Please cite previous publications and only draw conclusion based on previous studies.

Response: Thanks for the valuable suggestion. We now added relevant references (Highlighted in green) and rewritten the paragraph based on previous published studies. We have also included a graphical illustration of ES and CLA effects on bone health (Figure 5). We removed any statements not supported by previous publications.

We have added references to support “of electric stimulation on ERK, PKA signaling” We removed the statement “Electric stimulation may influence CLA-related biochemical pathways by reducing pro-inflammatory cytokines”

All changes are highlighted in green for your convenience.

Reviewer 3 Report

Comments and Suggestions for Authors

This manuscript aims to review the mechanisms, challenges, and innovative strategies for using conjugated linoleic acid (CLA) to enhance bone health.

  1. Section 2.2 is titled “Clinical Trial.” However, not all studies included in this section are clinical trials, such as reference 14.
  2. Table 2 includes 11 studies, seven of which indicate that CLA supplementation has no beneficial effect on bone health, particularly bone mineral density (BMD). This raises concerns about the effectiveness of CLA in promoting bone health, yet the authors have overlooked these negative findings.
  3. The authors do not provide evidence supporting the synergistic effect of electrical stimulation and CLA on bone health. In Section 4.1, the proposed synergy with electrical stimulation appears to be speculative and seems to align with the authors’ own assumptions rather than being supported by empirical evidence.

Author Response

Comments and Suggestions for Authors

This manuscript aims to review the mechanisms, challenges, and innovative strategies for using conjugated linoleic acid (CLA) to enhance bone health.

  1. Section 2.2 is titled “Clinical Trial.” However, not all studies included in this section are clinical trials, such as reference 14.

Response: Thank you for your valuable feedback. In response, we have removed the mentioned study as per your suggestion and have revised the section accordingly.

  1. Table 2 includes 11 studies, seven of which indicate that CLA supplementation has no beneficial effect on bone health, particularly bone mineral density (BMD). This raises concerns about the effectiveness of CLA in promoting bone health, yet the authors have overlooked these negative findings.

Response: Thank you for your valuable feedback. We have carefully reanalyzed each study and have revised Table 2 accordingly. We have ensured that the table now includes clinical trials that also demonstrate positive effects of CLA on bone health while maintaining a balanced perspective. We acknowledge that some studies show no effect, and many were inconclusive. To address this, our review includes strategies for enhancing CLA’s efficacy, such as nanoparticle delivery systems and electrical stimulation. These approaches have the potential to improve CLA’s effectiveness in promoting bone health. We aim to demonstrate in our review that these strategies provide a solution to the limitations observed in clinical studies.

  1. The authors do not provide evidence supporting the synergistic effect of electrical stimulation and CLA on bone health. In Section 4.1, the proposed synergy with electrical stimulation appears to be speculative and seems to align with the authors’ own assumptions rather than being supported by empirical evidence.

Response: Currently, no specific studies have investigated the combined effects of electrical stimulation (ES) and conjugated linoleic acid (CLA) supplementation on bone remodeling. However, as highlighted in the newly added Table 4, both ES and CLA share common mechanisms that promote bone regeneration. Based on this shared potential, we propose that the combination of ES and CLA may exert a synergistic effect, further enhancing bone remodeling. While numerous preclinical studies have demonstrated promising effects of CLA on bone health, clinical findings remain inconclusive—possibly due to issues related to drug delivery or poor bioavailability. Therefore, in this review, we recommend exploring nanoparticle-based drug delivery systems in conjunction with emerging ES technologies to optimize bone health outcomes. Nevertheless, further preclinical and clinical investigations are essential to validate the efficacy of this combinatorial approach.  

All changes are highlighted in green for your convenience.

Round 2

Reviewer 1 Report

Comments and Suggestions for Authors

The paper is better. The questions have answered by the authors except the references

Author Response

Reviewer 1 comment:

The paper is better. The questions have answered by the authors except the references.

Response:

Thank you very much for your positive comments and encouraging feedback. Regarding the references, we have made every effort to format them as closely as possible to the journal’s guidelines. However, the exact citation style is not available in the reference management software we are currently using, and manually adjusting all entries would be quite time-consuming and may introduce inconsistencies.

We hope MDPI team will kindly assist in converting the references to the exact required format. In our previous submission with MDPI, we received similar support, which was immensely helpful.

Reviewer 3 Report

Comments and Suggestions for Authors

Ok,  no more comments.

Author Response

We appreciate your guidance for improving our manuscript.